# Ternary Moral Empathy Model from the Perspective of Intersubjective Phenomenology

**DOI:** 10.3390/bs14090792

**Published:** 2024-09-09

**Authors:** Zhihui Zhao, Xiangzhen Ma

**Affiliations:** School of Humanities, Southeast University, Nanjing 211189, China; psydocseu@seu.edu.cn

**Keywords:** moral empathy, intersubjectivity, embodiment, lifeworld, ontogeny

## Abstract

The phenomenon of empathy is an intersubjective process of feeling and a particular form of intentionality. Moral empathy refers to a type of empathy that can trigger moral action, with the embodied intersubjectivity laying the foundation for its emergence. This paper attempts to propose a comprehensive theoretical model of moral empathy from the perspective of intersubjective phenomenology, which includes the following. (1) The moral dimension of perceptual empathy: at the subpersonal, unconscious, and perceptual–motor level, embodied empathic practices are essential for the formation of moral consciousness and the emergence of moral empathy. (2) The moral dimension of situational empathy: following the development of shared attention mechanisms, children can direct towards the intentional objects of others through embodied situational cues to perceive the psychological state of others and generate the moral empathy of “ought”, leading to dyadic morality that promotes cooperative behavior. (3) The moral dimension of narrative empathy: the narrative practices of moral empathy refer to the processes by which children could perceive and understand the moral situation of characters within an embodied narrative structure, subsequently generate prosocial motives such as empathic concern, and then accept the “objective” moral norms of the group consciousness embedded in the narrative.

## 1. Introduction

Empathy has become an increasingly important topic in various subfields of psychology, including cognitive neuroscience, developmental psychology, and moral psychology. Despite significant advances in the psychological study of empathy, the concept of empathy still lacks coherence and clarity [1]. Conclusions drawn from inconsistent concepts lack universal applicability, which poses a challenge to the development of the field of empathy. It is therefore necessary to clarify the essential nature of empathy.

As empathy is a multidimensional phenomenon and complex construct, numerous theoretical accounts of empathy have been proposed. Hoffman suggests that empathy must be a multi-determined response that can be aroused by five different modes, including three preverbal, automatic, and involuntary modes: mimicry, classical conditioning, and direct association; and two higher-order cognitive modes: mediated association and perspective taking. Hoffman’s theoretical framework for the development of empathy includes four stages: the stage of confused self–other differentiation; egocentric or quasi-egocentric empathy means that infants still confuse their own internal states with those of others and try to use helping strategies to reduce their own distress; veridical empathy means that toddlers can realize others’ independent inner states and empathize their actual distress more accurately; the last is the empathy stage, at which children can transcend the immediate situational cues and empathize with other’s general life conditions [2].

The Russian doll model of empathy consists of emotional contagion, sympathetic concern, and targeted helping, with the automatic state matching at the inner core based on the perception–action mechanism, and the more complex forms such as the perspective-taking ability at the outer layer, yet the outer layers still remain fundamentally connected to the inner core [3,4]. As cognitive empathy increases, infants progress beyond the basic state-matching stage. Decety contends that empathy mainly contains two components that rely on dissociable information-processing mechanisms: the emotional component involves emotion sharing through bottom-up processing, in which the shared neural circuits underlie the first-hand experience and the perception of other’s experience; the cognitive component involves self-regulation and modulation through top-down processing, in which emotion regulation, voluntary control, and intentions influence the empathic response [5,6]. The dual-process model of empathy attempts to integrate cognitive neuroscience and social psychology, synthesizing various theoretical notions and empirical findings into a fine conceptual analysis of empathy.

This review shows that the prevailing psychological models of empathy focus primarily on state matching and biological mechanisms, and are predominantly based on the Cartesian dualism of mind/body and subject/object. But in fact, the empathizer’s affective state is not isomorphic to the target’s state, as the empathizer does not have first-person access to others’ minds [7]. Current research on empathy is mainly conceptualized within the Theory of Mind (ToM) framework, which posits the capacity to attribute mental states to oneself and others, and which primarily encompasses simulation theory (ST) and theory-theory (TT). However, ToM encounters several theoretical challenges in effectively elucidating the nature of empathy. For instance, ST has drawn on neuroscience, particularly the resonance system and the mirror neurons system (MNs), to provide scientific support for a form of implicit simulation and an embodied simulation approach. Goldman suggests that the direct-matching hypothesis fits neatly with the simulation theory of emotion processing [8]. While there are several conceptual problems that contradict the theoretical assumptions of ST, namely that the subpersonal mirroring mechanisms lack an instrumental character and do not involve a pretense that requires a conscious distinction between the agent and the observer [9]. On the other hand, the TT perspective of empathy is not so much an experience as a knowledge of an unfamiliar experience, because it fails to capture the conscious experience of unfamiliar subjects. Moreover, TT is merely an attribution of non-propositional affective states in terms of propositional attitudes, which does not effectively explain the relationship between our theoretical understanding of emotions and our ability to experience affective states [10,11]. Indeed, empathy is essentially an experience of the other’s mind, distinguishing it from the first-person perspective of ST and the third-person perspective of TT. Thus, ToM has difficulty explaining the nature of empathic behavior because of its inadequate explanatory power.

Returning to the lifeworld (i.e., the everyday world in which we live pre-reflectively and experientially), however, we can find that in most interpersonal interactions and encounters between subjects, individuals can directly perceive the mental states of others because the empathic experience is given directly as an existence in the here and now, and it is a fundamental and intuitive fact that can be perceived in an embodied way. Given that the above models overlook the significance of contextualized situations, narrative structures, and the interactive nature of empathy, they fail to account for all that empathy entails. Therefore, we aim to propose a more comprehensive theory of empathy in terms of intersubjective phenomenology, by clarifying the nature of empathy and synthesizing diverse definitions and manifestations. An alternative phenomenological approach to empathy can employ the method of phenomenological reduction to provide a phenomenological description of the essential characteristics and intentional structures of empathic behavior. Additionally, considering the wealth of theoretical and empirical research from moral psychology, social neuroscience [12], and ethics that illustrates the close relationship between empathy and morality, we further explore the different dimensions of moral empathy from the perspective of intersubjective phenomenology.

## 2. Moral Empathy in Intersubjective Phenomenology

Contrary to earlier models, a phenomenological account can contribute to the conceptual clarification of empathy. We begin this section by clarifying the connection between empathy and intersubjectivity, and the embodied nature of empathy. We will then explore the complex relationship between empathy and morality and construct a three-dimensional structured model of moral empathy within an intersubjectivity framework.

### 2.1. Phenomenology of Empathy

In the phenomenological tradition, intersubjectivity can be understood simply as the relationship between subjects and the interdependence of subjects in the world. Merleau-Ponty’s intercorporeality advances and broadens Husserl’s understanding of intersubjectivity. That is, interactions between subjects are fundamentally rooted in intercorporeality, with the body playing a pivotal role in elucidating the potential and modes of realization of intersubjectivity. The properties of the “individual” or “individual self” are intricately linked to the body which is understood as a psycho-physical unity [13]. A full account of bodily experiences consists of a subjectively lived body (Leib) and a physical or objective body (Korper), and the subject is inherently an embodied entity capable of perceiving the mental states of others in the second person, which provides one with the means to recognize others as embodied subjects. In other words, second-person interactions are predominantly characterized by direct, practical, and non-inferential embodied contact and engagement, thereby rendering intersubjectivity as fundamentally embodied.

Gallagher categorizes the hierarchy of intersubjectivity according to the degree of intersubjective experience associated with bodily development. He posits three interrelated processes that are sufficient to develop the nuanced adult capacity to understand others: (1) primary intersubjectivity refers to the emergence of basic sensorimotor abilities in early infancy, enabling face-to-face dyadic interactions between mother and infant; (2) secondary intersubjectivity begins to develop at around 1 year of age and is primarily based on shared attention mechanisms or joint intentions, which describe the ability of two individuals to simultaneously focus their attention on the same object or event, facilitating situational engagement and cooperative behavior; (3) narrative competency typically develops between the ages of 2 and 4 and involves narrative practices that capture intersubjective interactions, motivations, and reasons [14,15]. In short, this positive account mainly involves intersubjective perception, pragmatically contextualized comprehension, and narrative competency.

According to phenomenologists, empathy represents a kind of quasi-perceptual awareness of another’s psychological state and experience of the other’s embodied mind. Husserl distinguishes different levels of empathy, the most primitive of which is the appresentation of the other body as constitution by coupling and the unconscious associative bonding based on the bodily similarity between self and other [16]. Thus, a phenomenological account of intersubjectivity and embodiment could contribute to a satisfactory explanation of empathy. Fuchs et al. propose an embodied affectivity model and a new concept called interaffectivity, which attempt to explain the complex connections between affectivity and embodiment [17]. Within the enactive approach, empathy emerges as a process from the continuous dynamic interactions between brain, body, and environment, and is considered as a basic process of affective response to the bodily presence of others through our direct bodily perception and sensation [18,19]. But in a phenomenological account, one can empathize not only with affective states, but also with the cognitive and conative experiences of others [16].

On the basis of these theoretical resources and the literature, we define empathy as an intentional act that can be directly directed towards the subjective affective and cognitive mental states of others through embodied expressive behavior and meaningful actions in a shared real-life situation. This definition distinguishes empathy from sympathy or compassion, as the latter is not necessarily directed at the intentional objects of another person, i.e., it can elicit responses without experiencing the emotional pain of others.

In conclusion, empathy is an intersubjective phenomenon, distinct from the egocentric emotional projections of paranoid subjectivity. It involves an intersubjective process of feeling and a basic pattern of intentionality (i.e., the aboutness of mental states), with intersubjectivity serving as the essential prerequisite for the emergence of empathy. Through empathy we can recognize the psychological states of others and gain self-knowledge by understanding how others perceive our experiences. Clarifying and elucidating the concept and nature of empathy is a fundamental aspect of intersubjective phenomenology.

### 2.2. Ternary Moral Empathy Model

Morality typically encompasses concepts of justice, fairness, and rights, as well as the value of other’s well-being. This article primarily focuses on the latter aspect. Scholars have synthesized developmental research on the development of moral judgment and behavior, positing that the human’s sociomoral core should be present from infancy, remains intact throughout one’s life, and constrains the influence of experiences and maturity in other domains on moral development [20]. In fact, the relationship between empathy and morality is complex, because empathy does not always lead to moral behavior. Excluding empathy types in the social cognitive sense, which lacks moral significance, the relationship between morality and empathy can be categorized into moral empathy and immoral empathy. Immoral empathy means that empathy deficits may contribute to violent and immoral behavior [21]. As for moral empathy, it is a type of empathy that can trigger moral action and generate prosocial behavior. Compared to other forms of moral action, the advantages of moral empathy are that empathy can be coupled with moral principles and affective motivation, illustrating the congruence between empathic emotion and moral evaluation [22].

Moreover, the relational nature of morality and the concrete encounters between children in their real life appear to be an important basis for understanding the value of others’ well-being [23]. Thus, the framework of interpersonal interactions and moral significance should be situated within the context of intersubjectivity. Given that children’s moral experiences are formed within relationships, it is crucial to adopt an intersubjective perspective to explain moral empathy. The interactions between children and theories of intersubjectivity offer an effective way to understand the early experience of moral empathy. Various forms of intersubjectivity provide the basis for moral empathy, and the embodied characteristics of intersubjectivity facilitate the generation of moral empathy. Therefore, moral empathy should be integrated into the overarching framework of intersubjectivity. Different experiences of moral empathy should be located at appropriate levels of intersubjectivity, leading to the development of a theoretical model of moral empathy that includes perceptual, situational, and narrative dimensions.

Specifically, a comprehensive theoretical system of moral empathy should be grasped from the subpersonal, personal, and socio-ethical levels, and the article on theoretical construction is divided into three sections. (1) The moral dimension of perceptual empathy: Primary embodiment originates in infancy, when newborns automatically mimic the facial expressions of others, representing a subpersonal, unconscious, perceptual–motor level of embodied empathy. This stage is pre-moral but essential for the development of moral consciousness and the emergence of moral empathy. (2) The moral dimension of situational empathy: Primary intersubjectivity is complemented and enhanced by secondary intersubjectivity. With the development of shared intentions, individuals can identify the emotional needs of others through embodied situational cues, thereby promoting their prosocial behavior. We aim to investigate the development of “second-person morality” through situational moral empathy. (3) The moral dimension of narrative empathy: Once children acquire the capabilities of primary and secondary intersubjectivity, tertiary embodied intersubjectivity necessitates mature narrative competency for complex intersubjective interactions and a nuanced understanding of others. At this stage, children are ready to comprehend the moral situations of characters and the objective moral norms within moral narrative structures.

Moral empathy is an important moral emotion and a valued capacity, but very few studies focus on its early ontogeny. This article presents a preliminary integration of the concept of moral empathy into the framework of intersubjectivity and offers an ontogenetic analysis of its structure and different dimensions. As presented in Figure 1, the rudimentary form of moral empathy emerges in early childhood, demonstrating both a specific development at each stage and a progression towards higher levels of intersubjectivity. These dimensions of moral empathy manifest themselves in primitive forms during early childhood and develop as psychological development and intersubjective capacity increase, progressing from primary to advanced levels of moral empathy. It is important to note that these moral empathic competencies of different levels of intersubjectivity are maintained and become more complex after adulthood.

## 3. The Moral Dimension of Perceptual Empathy

Primary intersubjectivity refers to the sense of shared experience and matching of others’ emotions and intentions, which develops in the context of dyadic, intimate, face-to-face interactions during infancy. To engage in communication, infants must be able to adapt their subjective control to the subjectivity of others [24]. Subjectivity here means that an individual has personal awareness and intentionality. Scholars generally believe that the differentiation between self and other requires reflective, conceptual self-awareness, typically evidenced by mirror self-recognition tasks [2]. Contrarily, an alternative perspective posits that a simpler, implicit form of self-awareness may be sufficient to experience the mental states of others. The literature on self-concept indicates that a basic, pre-reflective form of self-awareness is present from the beginning of life [25]. The development of bodily self-awareness in 9-month-old infants demonstrates the ability to differentiate between self and others, which is significantly correlated with empathic emotions towards others’ pain [26]. We propose that infants are capable of discriminating between the self and others prior to the formation of a true reflective, conceptual sense of self, implying that a rudimentary form of empathy is already present.

At the stage of primary intersubjectivity, perceptual empathy can be seen as a direct, practical (non-conceptual) perceptual experience of another’s psychological state, which is essentially embodied empathy. This fundamental form of empathy is pivotal in the development of moral empathy, serving as a cornerstone for children’s moral capacities in an ontogenetic sense. Briefly, in the first section, we suggest perceptual empathy relies on the perceptual experience of others’ bodily expressions and psychological states, thus laying the groundwork for the development of moral empathy and subsequent moral growth in children.

### 3.1. Embodied Empathic Practices of Perception-Action Coupling

In the pre-cognitive stage, infants’ empathy experiences are influenced by practical patterns of perception–action, in which infants’ primary bodily coordination activities can be seen as expressions of purpose and sensation. Trevarthen hypothesizes that infants are endowed with a brain system capable of responding adaptively in emotionally regulated interactions with supportive others; he refers to this innate ability and interpersonal activity as “primary intersubjectivity” [27]. Trevarthen emphasizes that infants are born with the capacity to understand others’ motives in the process of emotional, intentional, and meaningful negotiation, characterizing the early dyadic “proto-conversation” between infants and caregivers.

Intersubjective affective communication has a common innate basis related to perceptual, motor, and emotional intentional patterns of the body, which are essential for developing social cognition and empathy. From early life, infants display coordinated emotional states and synchronized rhythmic movements with their caregivers. The tendency to recognize, share, and respond to the emotional states of others is evident from birth, with infants inherently possessing the motivations and emotions necessary to sustain human intersubjectivity and demonstrating a highly coherent, rhythmic, and purposeful consciousness [28]. In Hoffman’s model of empathic development, the most basic pattern of empathic arousal is the primary circular reaction that occurs when an infant cries upon hearing another’s cry, involving emotional components rather than mere imitation. This reactive crying is thought to stem from an innate releasing mechanism, serving as both an early form and a precursor to more sophisticated empathic behaviors [29]. While this phenomenon does not constitute true empathy, it exemplifies a rudimentary empathic distress response.

In addition, newborns can discriminate some basic facial expressions [30], which is essential for emotional recognition and empathic responses. At around 2 months of age, early intersubjectivity is evident as infants engage in multimodal communication with their mothers, mirroring their vocal tones, gestures, and facial expressions, and the emergence of a sense of sharing experiences and interacting with others marks the first significant milestone in the early development of social cognition [31,32]. Between 2 and 3 months, infants frequently engage in emotional synchronization during play with their mothers, laying a crucial groundwork for empathy development [33]. This phenomenon, known as “proto-conversation”, refers to the spontaneous face-to-face interactions characterized by patterned and rhythmic vocal, facial, and gestural expressions. During these interactions, mothers and infants cooperate in an alternating, non-overlapping vocal pattern, with mothers speaking in short sentences and infants responding with cooing and murmuring, together creating a brief, dialogue-like joint performance [34].

Infants aged 5 to 7 months demonstrate sensitivity to emotions expressed in faces and voices [35]. By at least 7 months, infants are able to discriminate bodily emotions and have begun to acquire emotional knowledge for cross-modal matching of emotional expressions, recognizing that facial, bodily, and vocal expressions describe the same underlying emotion [36]. By 9 months, infants can perceive general body movements as meaningful and goal-directed actions [37]. These observations preliminarily suggest that infants respond positively to adults’ emotional responses within dyadic interpersonal interactions, indicating a basic form of bodily coupling and a primary sense of intersubjectivity.

Furthermore, empathic engagement is supported by the various physical abilities mentioned above. Emotional mimicry, which emerges from early face-to-face interactions, consistently responds to others’ facial expressions and represents an early, pre-cognitive form of empathy [38]. This pre-reflective emotional phenomenon is significant for intersubjectivity, demonstrating infants’ capacity to integrate expressive bodily movements with state regulation and their sensitivity to the structural consistency of emotional interaction timing and others’ emotional experiences. Additionally, the shared neural network representations between self and others serve as a neural correlation with empathy. The emotional processing of one’s self and empathy for others’ emotions activate overlapping brain regions, facilitating the understanding of others’ feelings [39]. The specific neurophysiological basis of emotional mimicry is that infants’ imitation of others’ facial expressions activates the MNs, then the action representation information generated is transmitted to the emotion-processing limbic system via the insula, thereby producing a similar emotional experience as the one being imitated and achieving basic empathy [40].

Empathy induced by mimicry may represent a special form of intersubjective mirroring, offering a subpersonal experiential explanation for Husserl’s view on the coupling of self and other, as well as Merleau-Ponty’s notions of chiasm and reversibility. MNs connect perception and action, empathic subjects and empathic objects, by transforming the visual perception of another’s body into the proprioceptive perception of the acting subject, providing the necessary conditions for empathic experiences. The neural-level representational equivalence between perception and action may underpin the behavioral-level representational equivalence between self and other, potentially deepening social relationships and providing a functional bridge between first-person and third-person perspectives [41,42]. The mirror neuron theory may elucidate the subpersonal mechanisms underlying empathy and serve as a beneficial experiential supplement to the phenomenology of empathy, with basic forms of empathy potentially explained by the activity of MNs. In short, the discovery of MNs allows the possibility to conceive intersubjectivity from a novel neuroscientific perspective [43]. In addition, we should also focus on more concrete notions or bodily factors, such as interoception, which has been shown as an association between the perception of internal corporeal states and empathy [44]. This focus will further deepen and expand the theoretical and empirical exploration of intersubjectivity, embodiment, and empathy.

These empirical findings and development evidence suggest that intersubjectivity is the source of empathy, and that parent–child interactions fundamentally reflect a mirroring relationship. In the early stages of intersubjectivity, infants develop a sense of shared experience and interaction with others, which manifests as the primordial “face-to-face” relationship of “I-Thou” in the complex, multidimensional structure of the lifeworld. Caregivers’ emotional mirroring of imitation and response to the infants’ emotional expressions plays an important role in the development of empathy in infants. Children of depressed mothers experience less emotional mirroring and demonstrate lower empathy levels compared to children raised in typical environments [45]. Poor intersubjective relationships may hinder early cognitive development, damage emotional expression, diminish sensitivity to others’ emotions, and impede the growth of empathic capacities, potentially leading to disordered behavioral characteristics in infants. The absence of intersubjectivity deprives individuals of the opportunity to develop prosocial behaviors, empathy, and moral judgment [31]. We therefore emphasize the socio-emotional nature of intersubjectivity and advocate for the prioritization of early companionship and socio-emotional relationships to foster empathic competency.

In brief, the empathic behavior at the primary intersubjective stage is rooted in the embodied practices of affective and sensorimotor processes associated with others’ bodily expressive actions. It is characterized by pre-reflective and non-conceptual attributes, rather than by theoretical inferences or simulations of emotional states. Even at this early stage, infants demonstrate sophisticated emotional-processing abilities and early empathic capacities. Interactions between infants and caregivers that involve resonance, attunement, or echoing of emotions and sensations are crucial for the development of empathy and moral cognition. Before exploring the moral implications of perceptual empathy, it is essential to describe the ontogeny of empathy, providing an experiential basis for the transition from the social-cognitive to the moral-valuative aspects of empathy. The complex and non-fundamental emotional resonances and moral evaluations in the social–cultural world emerge from the infant’s dynamic responses to the intersubjective sense of “being with others”.

### 3.2. The Emergence of Moral Empathy

The development of empathy in early childhood is a significant predictor of later moral development. Emotional empathy observed in preschoolers may forecast their future prosocial behaviors, with genuine concern for others’ suffering developing into heightened moral awareness [46]. Although children under 2 years of age typically do not form moral concepts, their emotional responses significantly influence behavioral decisions and play a crucial role in their socialization and the development of moral emotions.

During the first year of life, infants not only experience their own emotional pain, but also show empathic concern for others’ suffering and attempt to cognitively comprehend it [47]. Liddle et al. [48] refined methods to study early prosocial behavior, particularly employing the babies-in-groups paradigm to investigate infants’ empathic and prosocial responses. Their findings indicated that 6-month-old infants can distinguish between their own pain and that of others, self-regulate in response to others’ emotional distress, and engage in proto-conversations and reciprocal awareness, demonstrating intersubjective engagement and responses to others’ emotional states. This suggests that infants possess the capacity for empathic concern and prosocial responses within the first year of life. The perception of others’ suffering is thought to be of significant evolutionary importance for the development of moral and prosocial behaviors, such as helping, sharing, and comforting [49]. Early sensitivity to the experience of pain in others may serve as a foundation for the development of human empathy. Contrary to earlier small-scale studies, Davidov et al. [50] employed a large longitudinal sample of infants aged 3–18 months and observed moderate empathic concern in infants aged 3–6 months, and early empathic responses were predictive of prosocial behavior beyond 18 months. These findings indicate that infants can distinguish between their own pain and that of others from an early age, suggesting that empathic concern emerges earlier than previous theoretical assumptions about the development of empathy.

At this stage, perceptual empathy predominantly displays a natural, biologically driven prosocial inclination, characterized by synchronized bodily movements and coordinated socio-emotional expressions. Perceptual empathy is predominantly presented as a primitive form of emotional empathy with minimal socio-cognitive components and manifests as motor empathy, empathic mimicry/imitation, and non-inferential empathy. Perceptual empathy also includes the most primitive good–evil intentions. The capacity to evaluate others based on their prosocial and antisocial behaviors emerges within the first year of life [51]. Our most complex aspects of social–emotional life, interpersonal relationships, and interactions are likely to be constructed from these elementary components through motion perception and neural mechanisms [52]. The emergence and development of early empathic concern and moral evaluation is also based on the biological mechanisms of the physical body. In other words, perceptual intentionality in infancy includes empathic concern, preferences, and rudimentary moral senses towards others, with normative evaluations based on social interaction. Consequently, this form of empathy enables infants to exhibit an initial form of moral empathy.

In summary, primary intersubjectivity is the cradle of empathic emotion and moral awareness, with moral empathy originating from early interactions between infants and their caregivers. Specifically speaking, this section mainly discusses embodied empathic practices and the emergence of moral empathy that provides emotional support for children’s subsequent moral development. Empathic concern in the first year serves as an ontogenetic preparation for prosocial behavior after the second year and provides a basis for explaining and predicting moral empathy in complex social scenarios. Empirical findings clarify the nature of continuity in the development of the social–moral domain and demonstrate that early empathic differences between individuals can predict future social–moral behavior. Future research should focus on replicating these preliminary findings and further exploring the specific roles of biological, emotional, cognitive, and social factors in this developmental continuity [53].

However, primitive empathy alone is not sufficient for a full understanding of the psychological lives of others. It is essential to broaden our focus beyond face-to-face interactions and dyadic contexts. In the next section, studies of moral empathy of secondary intersubjectivity move into the social environment, where biological factors and social developmental conditions jointly influence the moral development of young children.

## 4. The Moral Dimension of Situational Empathy

Empathy, a highly flexible and adaptive process, is closely linked to specific environmental cues that influence its emotional, sensorimotor, and cognitive processing. Similarly, moral empathy is shaped by situational factors that provide a context for the emergence of prosocial and moral behavior. Understanding of the ethical circumstances of empathic objects is also highly dependent on these situational cues.

Moral research on situational empathy aims to induce agents to feel the emotional experiences of others and to stimulate individuals’ moral motives and behaviors, starting from concrete social situations in the lifeworld. Acting subjects are always embedded within embodied contexts where they actively and continuously engage in intersubjective affective interactions and direct their intentionality towards the mental states of others, thus generating and shaping intersubjective moral empathy. Empathic arousal is crucial for developing moral sensitivity, which in turn automatically represents social events with emotional value [54]. Descriptions of others’ moral situations and suffering cues highlight the moral salience of social events, and situational empathic arousal helps to increase subjects’ moral sensitivity by making them more aware of others’ moral demands.

In this section, we first describe the emergence of attention mechanisms and shared awareness towards the end of the first year, suggesting that infants can transfer perceptual empathy from dyadic to triadic contexts. This transfer suggests that dyadic interactions are foundational to triadic interactions, laying the groundwork for higher-order intersubjectivity and more complex forms of empathy. We then find that by the age of two, children show a clear tendency to cooperate with others and begin to consciously consider social norms [55]. During the stage of secondary intersubjectivity, the underlying situation always frames the meaning of intersubjective moral encounters.

### 4.1. The Formation of “We-Intentionality”

Phenomenologists claim that interpreting the behavior and emotional expressions of others typically requires a meaningful and practical context. Situational empathy operates implicitly and tacitly in a context in which passive empathy is rooted in perceiving actions, enabling an individual to perceive the pre-thematized presence of another in a pre-predictive manner [56]. All encounters take place within a situation that presupposes mutual understanding [57], and perceiving an individual as an embodied entity within this environment facilitates a profound understanding of their mental states. Dynamic social situations underpin intersubjective interactions and serve as an active structuring framework for our emotional engagement with the world [58]. In everyday social interactions, our perception of others’ emotions is context-dependent; thus, emotions should be perceived in a real-time, situational manner rather than statically and in isolation [59]. In highly structured contexts, individuals can grasp the meaningful unity of life experiences and the social milieu, thereby enhancing their ability to accurately understand others’ psychological states and engage in intersubjective affective exchange.

Typically, around the age of 1 year, infants begin to develop shared attention mechanisms, along with capabilities for pragmatic sharing and social referencing. This development enables them to equally link mother, infant, and object as participants in shared intentions and joint actions. Consequently, infants transition into the stage of secondary intersubjectivity with the differentiation and development of primary intersubjectivity. The new form of cooperative intersubjectivity, termed person–person–object awareness, in which children’s movements, gestures, and expressions are no longer isolated but begin to be embedded in contexts and environments, integrating with the world as a whole [15].

The second year is a critical period for the development of empathy, as well as self-recognition, self-awareness, and linguistic self-reference. The emergence of shared attention mechanisms and a sense of cooperation are significant indicators of the development of moral empathy. To overcome the constraints of perceptual empathy inherent in primary intersubjectivity, moral empathy necessitates additional situational cues or environmental contexts, enabling young children to develop a more sophisticated capacity for moral empathy by placing the moral context within a broader social, cultural, and historical background.

There is much more to be said about the development of empathy in secondary intersubjectivity, which primarily represents a significant variation as the relationship between subjects is strengthened and the “we” of the transpersonal entity is initially formed. Based on the second-person interaction of primary intersubjectivity, the individual first establishes a connection with others and the lifeworld, and the essentially we-centric orientation is further developed and consolidated [60]. The interaction scenarios in secondary intersubjectivity go beyond the emotional mimicry observed in earlier face-to-face interactions, where an individual first experiences a sense of dwelling in the world. Empathy emerges as a highly situational cognitive process, with its spectrum of associated feelings, cognitions, and actions being highly dependent on the appropriate handling of the contextual situation.

Objects and events that attract the attention of infants and others are often utilized as mediators or representations in intersubjective affective interactions, by which infants can directly understand the emotional experiences and intentions of others. Intersubjectivity not only involves being emotionally attuned to others to sustain social relationships but also promotes cooperation and joint action between infants towards shared goals [31,61]. Gallagher posits that our perception of objects typically includes an affective dimension, allowing young children to access shared intersubjective feelings by focusing their attention on specific objects with others. As individuals’ nuanced social interaction practices gradually enrich the pragmatic context of society, their interactions with others will evolve and expand into engagements with the world [62].

Cognitive empathy, a crucial indicator of shared attention and collective intentionality, necessitates an awareness of self–other differentiation as its developmental prerequisite. The establishment of this awareness facilitates the regulation and monitoring of emotional sharing [63], advancing beyond mere shared representations of self and other, and enabling precise assessment of one’s own and another’s emotional states. In a fully developed empathic experience, the individual must possess both a sense of agency and self-awareness, hallmarks of mature empathy [64]. Cognitive empathy, as an advanced form of empathy, begins its rapid development in infants aged 1–2 years [65]. Enhanced cognitive abilities allow individuals to recognize and understand others’ emotions and dilemmas, enabling them to express empathic concern accurately and effectively, thereby avoiding potential misunderstandings.

By the age of two, children exhibit enhanced self-regulation and cognitive components of empathy that allow them not only to empathize with victims, but also to act on these empathic feelings. As embodied space expands and cognitive empathy develops, a new form of intentionality emerges. This new intentionality, termed “we-intentionality”, arises from empathy and mutual attention and requires a unique reciprocal second-person perspective [16]. We-intentionality is a specific form of intentionality directed towards shared goals and intentions, analogous to shared intentionality or collective intentionality. This first-person plural perspective enables individuals to recognize others as equal agents, facilitating a shift from egocentric perspectives to a collective “we” orientation.

To conclude this subsection, we should note that prosocial behavior typically emerges after empathy, as it demands a higher degree of self-regulation. This behavior involves not only recognizing the distress of others and feeling concerned for them, but also coordinating emotions and actions to produce a goal-directed response. The more complex integration of emotion, cognition, and action that underlies prosocial behavior may be facilitated by advances in self-regulation and other abilities during the second year of life [66]. As young children’s cognitive control and perspective-taking abilities improve, they facilitate the distinction between self and other perspectives, enabling the formation of other-directed intentionality that leads to the emergence of genuine moral awareness and prosocial behavior.

### 4.2. Dynamic Coupling of Embodied Situation and Moral Empathy

During secondary intersubjectivity, empathy shifts from being self-focused to being other-focused, strengthening its association with prosocial behavior. The relationship between empathic emotions and morality is complex and significantly influenced by situational factors. In particular, the embodied nature of empathy situates an individual’s understanding of others’ moral encounters within a specific situational context. From an embodied perspective, the moral significance of a situation is re-evaluated and moral empathy is seen as the result of a dynamic coupling of body, brain, and environment. Consequently, the dynamic nature of moral practice is best understood through embodied situations in which individuals can enhance their moral empathy, prosocial motivations, and moral awareness.

In contrast to symbolic computation, which uses symbols or symbol systems to simplify and express complex concepts and plays a central role in traditional cognitive science, situational cognition does not require internal representations. Instead, it directly accesses the “affordances” of the environment, facilitating real-time adaptive action in complex environments [67]. The term “affordance” describes the process by which environmental attributes are related to organismic actions, specifically the potential meaning that objects provide. Embodied situational cognition is characterized as distributive/generative cognition arising from the interactive generative process between the acting subject and the surrounding world (including both the physical and social environment), rather than a passive internal psychological representation of external information. From an embodied perspective, situational cognition is seen as an adaptive process in which cognition dynamically regulates actions, rather than mechanically following the input–output procedures of symbolic representations in an abstract, non-embodied, invariant, and context-free manner.

Due to methodological constraints in psychological experiments, research often excludes “situation” as a controlled variable, resulting in a lack of ecological validity in moral psychology. Situational factors influencing empathy mainly include spatio-temporal frameworks and intergroup relations, where the former pertains to the impact of temporal and spatial distance on empathy, and the latter refers to the social distance between the empathizer and the empathic target. Hence, moral empathy requires that moral agents have the capacity for situational adaptation and maintain sensitivity to different moral situations in order to make morally appropriate decisions.

More importantly, the responsive attitude of empathic emotion inherently has a second-person structure embedded in the practices of moral interaction. Empathy is crucial to the second-person perspective, as it requires the ability to take the perspective of others [68]. Thus, natural second-person moral empathy, arising from secondary intersubjectivity, constitutes the primary normative attitude of human beings. We seek to explore the emergence of “second-person morality” through moral empathy influenced by situational factors. The first step towards modern morality is to demonstrate empathic concern for unfamiliar individuals, thereby cultivating a novel form of Smithian empathy in which individuals empathize with others on the basis of self–other equivalence and shared experience [68]. Such second-person moral empathy promotes a sense of responsibility and obligation towards cooperative partners and discourages harmful actions.

Evolutionary theory in developmental science provides strong support for the claim that the human capacity for moral empathy and moral evaluation is rooted in basic systems that evolved in the cooperative context of collective life. Others are always given to us in meaningful situations in which they are seen as a new center of reference, influencing our perspective on the world, thus highlighting the intrinsic link between empathy and social referencing [16]. This statement emphasizes the interrelationship between the experiences of others and the constitution of the shared world. For example, our human ancestors developed moral empathy as a natural response to observing the pain and suffering of their peers, which fostered empathic anger, empathic concern, and altruistic behavior, giving rise to primitive morality. During the second intersubjective, children show early forms of moral empathy, which is an ontogenetic development of moral reciprocity and can also be seen as pure second-person moral empathy that does not involve general cultural norms.

Empirical research indicated that 18-month-old toddlers rarely exhibit empathy in shared situations, with only 25% of children in this age group displaying empathetic behavior. In contrast, two-thirds of 24-month-old toddlers demonstrated concern and sympathy for others in distress, with 10% exhibiting high levels of emotional concern. Children’s ability to respond emotionally to the distress of others becomes more pronounced in the second year of life [69]. The increased proportion of toddlers showing empathic concern may be related to their improved understanding of others’ painful situations. Among toddlers aged 18 to 25 months, 40% began to show signs of empathic concern when faced with an adult in a negative situation, even in the absence of visible facial expressions [70]. This suggests that empathic concern can emerge as early as 2 years of age and does not require explicit emotional cues or emotionally distressing responses, but rather an understanding of the psychological state of others through shared attentional capacity and situational cues.

In this section, we contend that social situations in the lifeworld always provide meaning frames for intersubjective moral encounters and cooperative interactions that gradually enable young children to develop moral empathy. Second-person moral empathy at this stage represents the primitive socially normative attitude of humans and can also explain the emergence of dyadic morality. What is more, the emergence of triadic attention is often seen as fundamental to the comprehensive development of social cognition and social interaction. This phenomenon is pivotal in language acquisition, which in turn facilitates a more complex and sophisticated form of joint attention: sharing thoughts [16]. This paper highlights important developments in young children before the age of 2 that contribute to the emergence of narrative competency. That is, between 15 and 24 months, toddlers develop language skills that enhance their communicative abilities. Furthermore, from 18 to 24 months, they begin to exhibit capabilities in episodic and autobiographical memory [9]. As children’s cognitive and linguistic abilities develop, moral empathy in secondary intersubjectivity will evolve into a more advanced form in the stage of narrative competency.

## 5. The Moral Dimension of Narrative Empathy

In ontogeny, narrative competency is a developmental process that enables individuals to recognize and influence the thoughts and behaviors of others through narrative expressions with purposes, emotions, intentions, and interests. Children’s narrative competency begins to emerge with the further development of language proficiency and shared attention mechanisms in the ontogenetic process. During the narrative competency stage, they begin to understand others’ emotions and articulate normative expectations using their acquired language. Language skills serve as an ideal tool for interpersonal communication, and narrative as an advanced form of empathic arousal expands the range of empathic feelings in children.

Indeed, when considering narrative empathy for diverse experiences, simulation mechanisms that rely solely on an individual’s limited first-person perspective may not accurately match the attributional goal with the mental states of others. Only when we situate others’ actions within the narrative framework of shared experience and engage with their embodied actions and the rich social environments in which they inhabit, can others’ behaviors become comprehensible and meaningful. Through narrative, we acquire the ability to engage in intersubjective interactions and dialogues, which in turn enhances our capacity to empathize with others’ mental states and moral situations. This practice is termed “narrative empathy”, which is characterized by the same intersubjective nature as perceptual and situational empathy and employs storytelling as an intersubjective approach to enhancing understanding of others.

In this section, we first suggest that a well-designed moral narrative should accurately reflect children’s moral development and meet the demands of the lifeworld. We then argue for the construction of an embodied narrative framework that highlights lived bodily experiences and enables embodied narratives to scaffold moral empathy.

### 5.1. The “Lifeworld” as the Prototype of Empathic Narrative

Our socio-cultural lifeworld abounds with overt symbolic phenomena such as language, gestures, and images. As a tool for coordinating and synchronizing intersubjective interactions, language can be used to construct an intersubjective or shared social reality. The term “narrative” refers to the art of storytelling, characterized by the representation of events organized sequentially and causally to convey a unified theme. Narrative scripts are frameworks and outlines for the development of social events in the lifeworld, including behavioral sequences and plot arrangements. These scripts enhance our understanding of special groups and different moral events in society, thereby expanding the individuals’ capacity for empathy.

The developmental psychologist Piaget pioneered the use of the “dual story” as a methodological tool for studying the development of moral reasoning in young children. Building on Piaget’s framework, Kohlberg further refined these narratives, focusing specifically on moral dilemmas to emphasize the moral dimensions of these life stories. The primary aim of these narrative assessments was to measure the complexity of children’s moral structures and to delineate the stages of their moral reasoning development [71]. Eisenberg criticizes Kohlberg’s method for focusing primarily on the prohibitive aspects of children’s moral judgements, arguing that morality is multifaceted. Accordingly, after reflecting on Kohlberg’s theory, Eisenberg developed the prosocial moral theory based on the prosocial moral dilemma [72].

In conclusion, a well-designed moral story test should accurately reflect children’s moral development in real-life scenarios. The key to designing a moral dilemma story is that the context, plot, and arrangement should be consistent with young children’s actual moral experiences. Each story situation should explore at least one universal moral theme, such as moral obligation and responsibility, fairness and justice, or moral intentions and consequences. For example, fictional narratives such as novels need to be drawn from our shared experiential environment so that the “embodied interaction” between readers and story characters can potentially change or deepen the reader’s engagement with the lifeworld.

Moral narratives serve to transform abstract moral principles into tangible experiences of moral empathy, thus overcoming the constraints of the face-to-face interactions of primary intersubjectivity and the context-specific scenarios of secondary intersubjectivity. Moral empathy can be used to describe “objective” morality through moral narratives, which can significantly expand an individual’s moral imagination. By engaging with moral narratives, individuals can vicariously experience the moral dilemmas of others with the aim of enlightening moral consciousness and internalizing moral values. That is, through moral narratives we can achieve intersubjective emotional resonance and value feedback, and then internalize the institutionalized “objective” moral norms. At the level of narrative competency, individuals begin to move from a shared intentionality limited to dyadic interactions to a collective intentionality suitable for broader social and cultural cooperation [73]. In the context of a collective culture, dyadic morality, limited to specific situations, evolves into objective morality, characterized by collective intentionality.

The ideal role within traditional cultural customs is considered entirely objective, as it is widely understood what role an individual, as a member of “we”, should fulfill based on cultural commonalities [73]. Various roles and communities in the lifeworld have applicable objective moral standards and ethical norms, and moral subjects in the ethical entity are expected to follow prescribed scripts and ideal role expectations that meet the standards of conduct in moral situations. Subjects are expected to act morally in accordance with the situations and plots inherent in the moral narrative, and their actions are then evaluated against moral norms to determine whether they have adequately fulfilled the duties and responsibilities of their roles, marking the genesis of modern morality. Accordingly, it is essential in educational practice to construct meta-narratives and narrative schemes tailored to the receptive capacities of young children. Moral stories and moral actions are the primary ways in which young children internalize moral knowledge, so it is necessary to develop a model for cultivating narrative empathy that matches the development pattern of young children’s moral cognition. These narratives should accurately reflect children’s initial life experiences and evolve to become more complex, embodied, and life-oriented, thus promoting the internalization and generation of moral awareness in children. The tacit moral knowledge gained through these narratives plays a crucial role in our embodied understanding and reorganization of moral life experiences in the real world.

This subsection indicates that only narrative frameworks that meet the requirements of the lifeworld are vivid, complete, and complex, rather than being mere abstract, indoctrinated intellectual structures. Narrative frameworks are conducive to enriching the moral subject’s moral experience and guiding readers’ empathic engagement, thus serving as a potential approach to moral empathy.

### 5.2. Embodied Narratives as “Scaffolding” for Moral Empathy

The significance of narrative empathy in social cognition lies in its ability to provide us with accurate intentionality and to connect with broader contexts and environments. For example, it is hard for me to simulate aliens’ experiences with my cognitive resources, but it is possible to empathize with them when we frame their behavior within a narrative to understand their history or situation. This means that I should be open to the experiences and lives of others from a rich diversity of narratives, rather than drawing from my own narrow experience [9,74].

The situational empathy acquired in secondary intersubjectivity becomes an essential component of narrative competency, which is a form of offline empathy that transcends immediate context and stimulates moral imagination through narrative storytelling and plot arrangement. Narrative empathy typically emerges around the age of 2, as young children’s language comprehension and working memory capacity improve, and with the cultivation of early oral narrative patterns between children and caregivers. At this stage, narrative competency builds on the foundation of secondary intersubjectivity, providing more subtle and complex ways of constructing the meaning of others’ emotional experiences. Children gradually grasp narrative frameworks that incorporate specific storylines, pragmatic contexts, temporal structures, and socio-cultural norms, which is a narrative practice that captures intersubjective interactions, motivations, and reasons [9]. This enables young children to empathize with the mental states of others, thereby fostering a prosocial motivation to care for the well-being of others.

Young children’s engagement in the intersubjective emotional interactions between narrative subjects, story characters, and readers in fairy tales, along with the creative imagination stimulated by the storyline, prepares them for complex narrative empathy. In fact, infants are already engaged in the emotional narrative process through meaningful behavior and primitive language symbols. Infant semiosis is emotional rather than merely representational or referential, and the emotional grammar exhibited by parents and young infants during play reveals an embryonic form of narrative organization [75]. As children’s conceptual understanding and language abilities gradually develop, and as their personal life experiences increase, they begin to form a broader perception of others’ non-immediate situations (those heard about, remembered, or read about in books), leading to a more complex form of empathy.

The narrative subject, as an embodied entity, represents the integration of body and mind, rooted in self-experience embedded in specific situations, rather than an abstract self or fictional object detached from contexts. Our embodied experiences, perceptions, and actions precede the narrative awareness of the self. Indeed, narratives are constructed around the experiences of the embodied subject, highlighting the lived bodily experiences before transforming these experiences into narratives, and this type of narrative emphasizing embodied experiences is termed “embodied narratives” [76].

The structure of embodied narratives typically comprises both abstract and concrete concepts, which jointly promote embodied emotions and perceptual experiences for the reader. Abstract concepts are interpreted as image schemas stemming from sensorimotor experiences and are derived from concrete concepts through metaphorical extension, indicating a close relationship between the two types of concepts. Concrete and abstract concepts contain distinct types of information: experiential (sensory, motor, and affective) and linguistic, with concrete concepts having a higher proportion of sensorimotor information and abstract concepts emphasizing emotional information [77]. This notion of embodied semantic representation gains further support from functional magnetic resonance imaging experiments. These experiments indicate that concrete concepts are grounded in sensorimotor experiences, whereas abstract concepts predominantly engage emotional processing [78]. It can be posited that both concrete and abstract concepts manifest varying levels of embodiment. Scorolli suggests that the embodied nature of concepts may be closely linked to their mode of acquisition; concepts acquired through direct interaction with referents exhibit stronger embodied representations, whereas those primarily acquired through language display weaker or indirect embodied representations [79].

Turner suggests that the power of literature and metaphor underlies the daily thinking of ordinary people [80]. Language can enhance the moral implications of semantics through metaphorical representations. Lakoff and Johnson propose conceptual metaphor theory through the analysis of a large corpus, positing that our conceptual systems are inherently metaphorical, and we use metaphorical discourse to make sense of the world [81]. Metaphors are essentially rooted in embodied experience and are shaped in the interaction between the body and the environment. In offline narrative cognition, body-based perceptual representations and specific social contexts can still provide a causal account of cognition and action for moral narratives.

Moreover, our ability to empathize with the moral experiences of others relies on an embodied capacity termed “synesthesia”, which involves a range of synesthetic perceptions of bodily sensations across modalities. This phenomenon is also linked to our moral sense and forms the basis of our systems of “moral metaphors” and “moral rhetoric” [82]. When engaging with moral narratives, individuals activate perceptual processes related to the moral metaphors and concepts presented in the narrative, suggesting that the understanding of moral semantics is based on the activation of low-level perceptual–motor systems. Essentially, synesthesia involves the “subjective existence” dimension of “subject-object” interaction, characterized by the mutual integration and transmission of different sensations, thus creating an open system of interactive communication. In addition, it includes the “value existence” dimension of “subject-subject” interaction, where synesthesia facilitates emotional resonance between individuals and underscores the moral significance of value transmission and the construction of moral life [83].

In sum, this narrative section elaborates the influence of narrative empathy on moral development is evident as children move from “second-person morality” to “objective morality” of group consciousness in the lifeworld. By constructing embodied empathic narratives that are embedded with moral values and moral norms, the narrative subject can effectively guide children to foster appropriate emotional responses to the moral dilemmas faced by the story characters. It should also be noted that moral narrative practices are unlikely to evoke primary perceptual empathy and situational empathy if they are obscured by abstract reasoning. Merlin Donald’s theory of “theoretical culture” posits the existence of an external memory, a device that allows for the preservation and communication of knowledge independent of human beings [84]. Sonesson’s third embodiment also considers literal symbols as objective artifacts and perceptible carriers of information after the materialization of the human mind, allowing narrative competency to remain at a distance from primary intersubjectivity and secondary intersubjectivity. This prevents perceptual empathy from being hindered by abstract symbolic representation and top-down cognition, which could lead to empathy being manipulated by dehumanizing tactics or influenced by questionable ideologies [60]. Narrative empathy should incorporate rich embodied information, including perceptual and situational cues. The moral imagination stimulated by embodied empathic narratives can bridge the gap between primal moral sentiments and abstract moral principles, allowing fictional narratives to scaffold social understanding and empathy [85] and function as a technology of the lifeworld [86].

## 6. Conclusions

From an intersubjective phenomenological perspective, children’s empathic concern and empathy-based moral behavior are conditioned by intersubjectivity, i.e., moral empathy is best addressed within intersubjective relations between subjects in the lifeworld. Drawing on Gallagher’s theory of intersubjectivity and integrating empirical findings from developmental psychology, cognitive science, and moral psychology, we propose a theoretical model of embodied moral empathy that includes three dimensions: perceptual moral empathy, situational moral empathy, and narrative moral empathy. This model aims to provide a philosophical foundation and a unified explanatory framework for exploring the relationship between empathy and morality. In addition to the perceptual factors that have been the focus of academic research, this article also highlights the influence of situational and narrative factors on empathy through the embodied intersubjectivity approach, which is beneficial for deepening and broadening the theoretical discussion on empathy. Future empirical studies should test and verify the hypotheses of the proposed phenomenological model of empathy.

We also argue that the integration of phenomenological theories into empirical research on empathy could yield more universally applicable findings and significantly enhance interdisciplinary empirical studies that are closely aligned with the complexities of the lifeworld. Phenomenological theory provides valuable theoretical guidance for empirical psychological research. In particular, the phenomenological approach could mitigate the risk inherent in the scientific method of pursuing overly precise, narrow operational definitions that may overlook the essential characteristics of empathy as observed in real-life situations. To further deepen our understanding of the nature of empathy, it is essential to explore the naturalized phenomenological approach that integrates phenomenology with cognitive science in empathy research. Recently, there has been a notable emergence of empirical and theoretical articles guided by phenomenological methods in the study of empathy. For example, some researchers provide a comprehensive view of empathy through an experimental phenomenological approach within an integrative theoretical and methodological framework that emphasizes the integration of intercorporeal interactions and phenomenological attributes, thereby enhancing our comprehension of the complexity of empathy by bridging the gap between laboratory-based research and real-life empathic experiences [19,44].

Finally, this article presents a preliminary exploration of moral empathy from a phenomenological perspective. We anticipate that future research will increasingly emphasize the moral value and significance of empathy from intersubjective or embodied perspectives, promoting active interdisciplinary integration and collaboration between phenomenology, ethics, and psychology.

## Figures and Tables

**Figure 1 behavsci-14-00792-f001:**
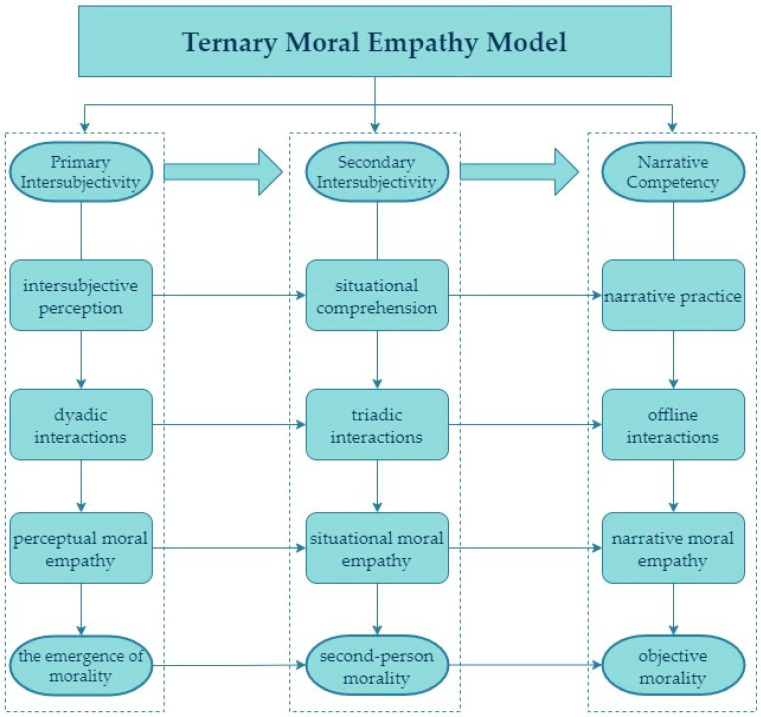
Ternary moral empathy model from the perspective of intersubjective phenomenology.

## Data Availability

No new data were created or analyzed in this study. Data sharing is not applicable to this article.

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
