# Peer review of "Ternary Moral Empathy Model from the Perspective of Intersubjective Phenomenology"

_behavsci, 2024, doi:10.3390/bs14090792_

Round 1

Reviewer 1 Report

Comments and Suggestions for Authors

The paper proposes a comprehensive theoretical model of moral empathy from the perspective of intersubjective phenomenology. It includes three dimensions: perceptual empathy, which focuses on unconscious, embodied practices; situational empathy, which involves shared attention and intentions in children; and narrative empathy, which relates to understanding moral situations through stories. While the model is detailed, several aspects require closer review and clarification to ensure its robustness.

One aspect that concerns me about the manuscript is the organization of the paper. The paper has several sections and subsections with themes that can be very similar, making it difficult to understand how the discursive theme of the paper evolves. I suggest adding a more extensive text at the end of the introduction, outlining the different themes that will be covered and the discursive evolution of the work. Additionally, at the beginning of each section, it should be explicitly stated what will be discussed throughout the argument, and at the end of the section, a conclusion should summarize the most relevant aspects that will help generate the integrative model. This will provide clarity and coherence to the structure of the paper, making it easier for the reader to follow the development of the authors' arguments.

“ It is widely accepted in psychology that empathy is the ability to perceive and under-stand another person's emotions and to respond in a certain way.”

Contrary to what is stated in this sentence, empathy has multiple and diverse conceptualizations. It is important to make this explicit first. Once done, provide a clear definition that will guide the manuscript, citing the appropriate reference.

Hall, J. A., & Schwartz, R. (2019). Empathy present and future. The Journal of Social Psychology, 159(3), 225–243. https://doi.org/10.1080/00224545.2018.1477442

I do not clearly recognize a distinction between the concepts of empathy, compassion, and morality in the text. I suggest that the authors improve this aspect by explicitly defining and differentiating these terms to enhance the clarity and depth of the discussion.

The authors present a theoretical model of embodied moral empathy, but it remains very blurred and unclear in the paper. I suggest several changes to enhance their work at the end of the manuscript. First, include a specific section within the paper where the theoretical model and the relationships between its concepts are clearly proposed. Second, add a figure that clearly presents the theoretical relationships of their proposal. Finally, present the central elements of innovation and integration of the proposal in relation to the current knowledge in embodied phenomenology.

The paper, quite appropriately in my view, conceptualizes empathy and morality as embodied processes. However, whenever the relationship between two individuals is discussed, the term "intersubjective" is used, reducing consciousness to a merely psychic event. I suggest the authors consider terms that are more faithful to this inter-corporal concept. Some authors use terms like "inter-corporeality," "inter-affectivity," and others.

While the theme of the body is present throughout the work, there is a lack of deeper exploration of concrete elements that can be understood within the body in the paper. For example, some authors propose that the sensorimotor system is essential for empathy, while others, without being mutually exclusive, attribute an essential aspect of empathy to interoceptive elements of the body. I suggest the authors delve deeper into these aspects to provide a more comprehensive understanding of the role of the body in empathy and morality.

While the authors present relevant literature to understand empathy as an intercorporeal and phenomenological process, I suggest integrating works that have previously addressed this topic, as it could significantly contribute to and strengthen their proposal

Colombetti, G. (2014). The feeling body: Affective science meets the enactive mind.

London: MIT Press.

Fuchs, T., and Koch, S. C. (2014). Embodied affectivity: On moving and being

moved. Front. Psychol. 5:508. doi: 10.3389/fpsyg.2014.00508

Troncoso, A., Soto, V., Gomila, A., & Martínez-Pernía, D. (2023). Moving beyond the lab: investigating empathy through the Empirical 5E approach. Frontiers in Psychology14, 1119469.

There are other relevant empirical works in the fields of neuroscience, phenomenology, and enaction that should be integrated into the argumentation of the paper. Introducing these studies would provide a more comprehensive foundation and strengthen the overall argument.

Martínez-Pernía, D., Cea, I., Troncoso, A., Blanco, K., Calderón Vergara, J., Baquedano, C., ... & Vergara, M. (2023). “I am feeling tension in my whole body”: An experimental phenomenological study of empathy for pain. Frontiers in Psychology13, 999227.

Gallese, V. (2014). Bodily selves in relation: Embodied simulation as second-person perspective on intersubjectivity. Philosophical Transactions of the Royal Society B: Biological Sciences, 369(1644), 20130177. https://doi.org/10.1098/rstb.2013.0177

Reviewer 2 Report

Comments and Suggestions for Authors

I find the paper interesting and informative, although the main constructive results are not too impressive.

But I have some reservations:

1. Introduction is not readable as it is, except if you know all the material beforehand. Perhaps the Authors want to delete it.

2. The Authors must specify what they mean by morality. Now, this concept's usage is fetishistic. What is moral empathy compared with other types of empathy?

3. The following (p. 3) does not sound right: "Therefore,
empathy can be viewed as a specific form of intentionality that relies on the expressive behavior and meaningful actions of the body directed towards the intentional objects of others." It seems to mean something like this: Empathy is a form of representation of expressive behavior and meaningful action directed towards other people's object of thought. In other words, I don't understand it.

4. It is not easy to distinguish between the Author's own theory development and reading of research literature.

5. The Authors load the text with unexplained terms and concepts, like we-intention. They tend to rush forward, which should not be necessary. For instance, this sentence is problematic in many ways (p.9): "Unlike symbolic computation, situational cognition does not necessitate internal
representations and can directly access the 'affordances' of the environment, enabling real-time adaptive action in complex environments [59]." The reader need a backgound explanation.

6. Consider using some examples and brief narratives to help the reader.

After a main rewrite this paper is publisable.

Round 2

Reviewer 1 Report

Comments and Suggestions for Authors

I accept the paper

Reviewer 2 Report

Comments and Suggestions for Authors

NO further comments-